

# Review Article: Validation of flood risk models: current practice and innovations

Daniela Molinari[1], Karin De Bruijn[2], Jessica Castillo[3], Giuseppe T. Aronica[4], Laurens M. Bouwer[2]

[1] Department of Civil and Environmental Engineering, Politecnico di Milano, Milan, 20133, Italy
[2] Deltares, Delft, 2629-HD, The Netherlands
[3] Research Institute on Water and Environmental Engineering, Universitat Politècnica de València, Valencia, 46022, Spain
[4] Department of Engineering, University of Messina, Messina, 98166, Italy

*Correspondence to:* Daniela Molinari (daniela.molinari@polimi.it)

**Abstract.** Although often neglected, model validation is a key topic in flood risk analysis, as flood risk estimates are characterised by significant levels of uncertainty. In this paper, we discuss the state of art of flood risk models validation, as concluded from the discussion among more than 50 experts at two main scientific events. The events aimed at identifying

policy and research recommendations towards promoting more common practice of validation, and an improvement of flood risk models reliability. We pay specific attention to the different components of the risk modelling chain (i.e. flood hazard, defence failure and flood damage analysis) as well as to their role into risk estimates, to highlight specificities and commonalities with respect to implemented techniques and research needs. The main conclusions from this review can be summarised as the need of higher quality data to perform validation and of benchmark solutions to be followed in different

contexts, along with a greater involvement of end-users in the debate on flood risk models validation.

## 1 Introduction

Model validation is a key topic in flood risk analysis, as flood risk assessments are characterised by significant levels of uncertainty (Handmer, 2003; Merz et al., 2010; Jongman et al., 2012); however, very few studies pay specific attention to the validation of flood risk estimates. Since large investments for flood risk mitigation are made based on flood risk estimates and

maps, decision makers must be aware of the limitations of accuracy of risk analysis outcomes. Also, in the insurance industry premiums are based on loss estimates from catastrophe models. Validation of risk outcomes is therefore crucial in order to gain confidence in the decisions made.

On 9 and 10 December, 2014, an expert workshop was held in Delft, The Netherlands, to discuss the topic of validation of flood risk models. The meeting was held under the auspices of the European Geosciences Union (EGU), and the Panta Rhei

Working Group "Changes in Flood Risk" of the International Association of Hydrological Sciences (IAHS). On that occasion, over 50 experts from 10 countries discussed model validation with respect to all the elements of flood risk analysis, namely



(1) hazard and inundation analysis and modelling, (2) failure and reliability modelling of flood defences, (3) flood damage modelling, and (4) the integration of hazards, failures, and damage to compute and evaluate risk.

In continuity with the workshop, a special session was organised at the Third European Conference on Flood Risk Management (FloodRisk 2016) in Lyon, France, on 19 October 2016. The session brought together scientists, policy makers and

representatives from the insurance sector to discuss the key challenges in flood hazard estimation and risk assessment, with a focus on the need to pay more attention to model validation.

This paper summarises the findings from these two events and identifies policy and research recommendations towards more common and recent practices of flood risk models validation. These recommendations may contribute to the overarching aim of improving the reliability and usability of results from flood risk analyses.

In the paper the following questions are addressed, as concluded from the discussions at the two events: (i) what is validation in flood risk modelling and why it is required? (ii) what techniques for validation are available for flood hazard, defence failure, and damage models? and (iii) which are the research and policy priorities in order to improve the reliability of present flood risk estimates?

## 2 What is validation in flood risk modelling

Despite the fact that validation and verification are quite often used interchangeably in engineering sciences, these terms refer to two different processes. The main difference is the way in which a product (or a service) is tested. According to ISO 9000 standard (www.iso.org), for example, verification refers to the confirmation that a product meets identified specifications, while validation concerns the evaluation that a product appropriately meets its design function or the intended use.

From the point of view of flood risk management, the compliance of a flood risk assessment with specific requirements is of

course important but, above all, it is to assure that such an assessment meets the needs of relevant stakeholders, stressing the key role of flood model validation.

Policy makers, for example, need validation to assess whether risk models' results are sufficiently accurate to take confident decisions. A key issue is the selection of the risk mitigation strategy to be implemented in a certain area at risk, among some promising alternatives like the construction of a levee to protect the area, the relocation of items in the area, or the

implementation of a flood early warning system to provide advice to residents. With respect to this, even in the presence of uncertain risk estimates, confident decisions require, at minimum, that the ranking of alternatives is certain (Molinari et al., 2016; Morales-Torres et al., 2016)

From another point of view, insurance companies need to validate risk models as their inaccuracy could imply wrong premium settings and loss of profitability. Less uncertainty is preferred, but knowing the uncertainty range better is already a great step

forward, since that enables insurers to more accurately assess risk premiums or, at least, to include uncertainty into "extra" premiums. From this perspective, it is very important that models implemented by insurance companies are consistent with some benchmarks.



At last, researchers validate risk models to gain trust in them, to understand what they may conclude from them and what not, and finally to identify the efforts required for their improvement.

Importantly, from any perspective, validation of risk models implies determining that output of the models has sufficient accuracy for the intended purpose. Validation is thus related to the decision for which the risk outcomes are to be used (Sayers et al., 2016).

Three main techniques are usually applied which allow validating risk models as well as components of them (e.g. the hazard model, the flood defence model, and, the damage model). The choice of the technique to be implemented depends mainly on the availability of data for validation.

The first technique consists of the comparison between model results and observed data. From this perspective, model validation is strongly linked with model calibration, that is the adjustment of model parameters, assumptions or equations to optimize the agreement between observed data and model's predictions. Still, validation and calibration are two distinct processes that should be taken at two different moments of models' development, respectively for the definition of the model structure and to test model usability.

Comparison with observed data requires first that the modelled quantity is measurable (risk, for example, cannot be observed on the field), and second that measurements of real data are actually available. In the case that observed data do not exist, the second technique of benchmarking with other (validated) models is an option. As third option, even in the absence of validated models to refer, falsification through expert knowledge and expert judgement can be applied which implies comparing model results with experts' expectations.

The state of implementation of the different techniques varies for the different risk models components. These are discussed in the next section. For each component, an overview is provided on (i) the main objective of model validation, (ii) data required for model validation, (iii) common validation techniques and (iv) new developments/research projects and activities to enhance validation and the most important challenges ahead.

## 3 State of art of flood risk validation

### 3.1 Hazard models

"Flood hazard" refers to the likelihood and the features of the damaging physical event in a particular location, such as the extent of the flooded area, the flood depths, the flow velocity, the duration of flooding, the water level rise rate, the concentration of sediments or other transported materials, and the pollution load of the water. Accordingly, validation of flood hazard models aims at evaluating how reliably these models estimate the probability and the characteristics of a flood event.

Validation techniques and challenges depend on the nature of the flood (e.g. riverine vs. flash floods, coastal, urban or groundwater floods) since for each flood type the most relevant parameters may be different. For example, water velocity is a crucial parameter for flash floods, while it may be not relevant for the prediction and management of slow-rising fluvial floods; validation of fluvial floods therefore usually concentrates on flood extents and water depths while validation of predicted water



velocities is critical in the estimation of flash floods. However, there are also general considerations which are valid for all flood types. These are discussed below.

Assessing the probability or likelihood of extreme flooding events is challenging. Often, validation is not possible, although extreme value analysis starts from recorded water levels or flow discharges. The uncertainty in prediction of probabilities is

very large for very extreme events which lie far beyond the highest recorded value, but smaller for more regular events. In any case, in absence of validation, uncertainty assessment is crucial.

Within the various methods available to estimate magnitudes and probabilities of extreme flood events and hydrologic loads (as streamflow-based statistical analysis or rainfall-based methods with statistical analysis on the generated runoff), the uncertainty is usually addressed by adopting different flow hydrographs as upstream boundary conditions or by doing Monte

Carlo simulations (e.g. Aronica et al., 2012; Domeneghetti et al. 2013,).

Next to probabilities, the  hazard  parameters inundation depth, flow velocity and flood extension are usually required for flood risk management decisions and therefore validation efforts concentrate on them (see, for instance, Thieken et al., 2005; Elmer et al., 2010; Merz et al., 2010; Merz et al. 2013; and section 3.3).

In this regard, there is little flood hazard data available since floods are rare and most of required data are not recorded after

floods (Aronica et al., 2002; Bates 2004): maximum water levels in water bodies are often quite reliable, but inundation depths and flow velocities are difficult to measure and often not recorded, and the extension of the flooded area maybe unknown for small/local events (Ballio et al., 2015). Accordingly, validation of flood hazard models is still underdeveloped in contrast to other models like e.g. hydrological models (Aronica et al., 2002; Bates, 2004).

Still, validation of flood hazard models is strongly recommended as they are affected by many sources of uncertainty: in

particular, the topographic description and the roughness parameters contribute significantly to the uncertainty in the modelled inundation extent and flow characteristics (Pappenberg et al., 2005; Papaioannou et al., 2016; Papaioannou et al., 2017)

In fact, new opportunities have arisen to better collect and organize event data, which support flood hazard model validation; these include: crowd sourcing (GPS tracks on smartphones), using satellite data (flooded area, waves), measuring with drones/local airplanes, studying high water marks, doing interviews with inhabitants, and using security camera footage (Di

Baldassarre et al., 2009; Bates, 2004; Schnebele et al., 2014; Gaume and Borga, 2008). Such techniques can also be exploited to derive information on other variables which are usually not recorded like sediments load and water level rise rate.

For validation in the absence of data, common sense can be used as a yardstick. For instance, discussing model results with experts (including those who know the system well) is critical, as they can spot peculiarities or inconsistencies in model results.

To handle with the problem of lack of validation, recent studies advocate the use of probabilistic instead of deterministic

approaches to flood hazard modelling, for two main reasons: (1) the uncertainty in hydrologic/hydraulic modelling process cannot be neglected; (2) probabilistic modelling approaches offer a way to address that uncertainty and to present it to flood risk managers in probabilistic flood inundation (e.g. Romanowicz and Beven, 2003; Di Baldassarre et al.; 2010, Alfonso et al., 2016; Candela and Aronica, 2017; Papaioannou et al., 2017).





In recent years, examples of frameworks for validating probabilistic flood models have been developed (Beven et al., 2011; EA, 2012). However, validating probabilistic model results for rare flood events and defence conditions, such as breaching, remains challenging.

## 3.2 Defence failure models

Defence failure models are needed to estimate the response of defence systems for different loading scenarios, and include engineering estimation, analysis of historical events, event and decision tree modelling, or Monte-Carlo simulations. Most dam and levee breach analyses aim also at predicting flooding conditions and resulting consequences downstream (Wahl, 2010).

The key role of defence failure models in the risk modelling chain is often neglected; nevertheless, their inclusion and validation is strongly recommended as they may significantly influence risk estimates. For example, the National Flood Risk

Assessment (NaFRA) research in the UK revealed that the uncertainty in the fragility curves, which are often used for representing the performance of flood defences, can propagated throughout the assessed risk having an impact of between 0.5 to 2 times the central estimate of the expected annual damage (EA, 2002).

Flood defence failure models are based on reliability analysis of defence systems, considering potential failure modes and correlations. Examples can be found (Steenbergen et al., 2004) and are commonly based on subdividing the flood defence

system into several parts (components) and the consideration of different failure mechanisms. Failure probabilities of each component and per mechanism are obtained (e.g. using numerical integration, Monte Carlo simulations, first or second order reliability methods, etc.) and combined at system scale.

Consequently, the aim of validating flood defence failure models is to identify key driving forces and characterize failure mechanisms (i.e. sliding, overtopping, stability, hydraulic soil failure, etc.).

Barriers to validation of flood defence failure models are many. Lack of data on evolution and progress of real failure cases is the first one. Indeed, although a few examples of historical flood defence failure events, levee failure databases and statistics can be found in the literature (e.g. Nagy and Toth, 2005; Ranzi et al., 2013), historical data on flood defence failures are not enough for fully characterizing all potential failure mechanisms and their corresponding initiating and progress events that lead to flood defence failure.

A second problem relates to the lack of results from experimental and laboratory tests from full-scale models. In fact, validation through full-scale testing has been conducted only in recent years (e.g. van Beek et al., 2010) and is now on-going. As an example, the IJkdijk program in the Netherlands can be quoted, which included a real (long-standing) levee system to test both failure and breach models.

At last, uncertainty on failure mechanisms and properties of materials prevents model validation, in spite of the extensive

research on geotechnical failure mechanisms of flood defences (e.g. dams or levees) such as instability or piping. Uncertainties on failure parameters (breach height, width, breach formation time and development, and peak outflow equations) should be further analysed (e.g. Froehlich, 2008; McCann and Paxson, 2016), evaluating their impact on risk results, along with other factors such as the length-effect or cascading failures (e.g. Schweckendiek et al., 2014).



Recent initiatives on improving defence systems analysis include, for example, actions within the SAFElevee project (TU Delft, the Netherlands) which aims to improve the reliability of flood defence systems by increasing understanding of their failure mechanisms, and the development of a cooperative knowledge platform on levee safety, which is also a basis for future research and validation of models.

In recent years, advances have been also made in how to approach the comparison of probabilistic outputs from defence performance analysis, focusing on probabilistic risk models (Sayers et al., 2016).

### 3.3 Damage models

The aim of damage models validation is to assess whether the models are able of reliably estimating the expected damage for a certain area (e.g. a municipality, a region) and for a given flood event (Merz et al., 2010).

From this perspective, flood damage models validation is hardly performed, the main constraint being the limited availability of high quality (damage) data (Merz et al., 2010; Jongman et al., 2012; Meyer et al., 2013; Molinari et al., 2014); availability that further decreases whit the increase of the resolution at which damage models work (de Moel et al., 2015) and when indirect and intangible damage is considered.

In fact, flood damage model validation requires huge information including: (i) observed damage to the different exposed

items (e.g. residential buildings, industrial buildings, farms, roads, etc.), preferably both in physical/quantitative and monetary terms, (ii) the spatial distribution of hazard variables, and (ii) the vulnerability of affected items; nonetheless, these data are required at a level of detail that can be even sub-local, e.g. when damage to different building subcomponents is assessed. Such information is usually not available, in the sense that only some of required data are known. Accordingly, validation in terms of comparison between model results and real data is difficult to implement; some examples can be found in Thieken et al.

(2008), Apel et al. (2009), Wuensch et al. (2009), Elmer et al. (2010), Seifert et al. (2010), and Dottori et al. (2016).

When historical flood data are not available, alternative methods are exploited. The most common one consists in the comparison among several damage models (see, for example, Ding et al., 2008; Bubeck et al., 2011; Dottori et al., 2016; Wagenaar et al. 2016). The difference in outcomes between different damage models does not result in conclusions on the precise damage figure, but provides insight in the uncertainty band width (Wagenaar et al., 2016). This uncertainty is larger

for local events in which only a few objects are affected, than in large-scale events which affect many objects because errors are partially cancelled out. The uncertainty is also larger for events with small water depths than for events causing large water depths. In the latter case the uncertainty on the reaction of citizens which may or may not effectively reduce impacts is smaller: it is more difficult to reduce flood impacts when water depths are larger. A new direction is the development of multi-variable damage models, using multiple hazard and exposure variables, by which the accuracy of the flood damage models is improved

(Merz et al., 2013).

At last, the use of expert knowledge can be exploited to check the reliability of model outcomes, in absence of data and models for validation (Dias et al., in press).



Next to the lack of observed data, also the strong context specific character of damage models offers a challenge in damage model validation. This makes transfer of models to other areas or time periods uncertain. Some, but only limited research is available on model transferability (see e.g. Cammerer et al., 2013; Molinari et al., 2014; Schroter et al., 2014; Wagenaar et al., 2016). Unfortunately, damage models are often applied in different spatial contexts without any validation.

Validation is critically needed as damage models, in the entire chain of flood risk analysis, are in many cases one of the most uncertain (Apel et al., 2009; de Moel et al., 2011), i.e. typically in areas without protection infrastructure.

Therefore, more coordinated efforts must be made to collect and share high quality damage data, through standardized data collection campaigns. Given the huge effort required by data collection, it is sensible that collected data regard not only damage itself, but also its explicative variables (Molinari et al., 2014). If data would be better collected and shared, we could in a near

future have thousands of comparable damage records and many data sets to derive and validate damage models. Data collection should be promoted at the level of individual flood affected items, in order to gain maximum information from the collection campaigns.

It must be acknowledged that the weakness in data availability has not gone unnoticed in science and constantly more efforts to collect flood damage data and to develop standardised methods are demanded by many authors and organisations (Elmer

2012); see, for example, Ramirez et al. (1988), Mileti (1999), NRC (1999), Guha-Sapir and Below (2002), WHO (2002), Yeo (2002), Rose (2004), Dilley et al. (2005), Downton and Pielke (2005), Handmer et al. (2005), UNISDR (2007), Merz et al. (2010),  Meyer et al. (2013), Molinari et al. (2014) and UNISDR (2015).

Fortunately, several attempts to improve data collection and availability have been implemented already (see Molinari et al. 2017, for a review): standardised procedures have been developed to collect data by means of telephone interviews (Thieken

et al., 2005; Kreibich et al., 2007; Kienzler et al., 2015, Thieken et al., 2017) or field surveys (Molinari et al., 2014; Ballio et al., 2015; Berni et al., 2017; King and Gurtner, 2017) after flood events. Furthermore, other possibilities like crowd sourcing or volunteered information approaches, data collection via dedicated web-sites, analysing information from social networks and similar have been explored (see e.g. Cervone et al., 2016; Frigerio et al., 2017; Roberts and Doyle, 2017). Another option is to verify the availability of useful data in the insurance sector. Still, these data are not usually available for the research so

that a greater commitment of the insurance sector into the problem at stake is desirable.

Also, efforts are in place to develop tools to manage and report collected data in a standardised way, in order to facilitate the comparison among different flood events and different areas (Menoni et al, 2016; Szoenyi et al., 2017). Future research should address the harmonization of data collection and reporting, in order to make the widest use of these data. In this respect, the initiative launched by the European Commission is significant (De Groeve et al., 2013; De Groeve et al., 2014; EU Commission

expert working group, 2015).

Future research could also include physical experiments to analyse flood damaging processes and to compensate for the lack of observed data. Still this approach can be adopted only at the micro scale and to analyse direct physical damage (see. e.g., Xiao and Li, 2013, Liang et al. 2016).





### 3.4 Integration of risk components

In the previous sections, validation of hazard and damage figures has been discussed. However, in flood risk analysis also the combination of both must be validated in order to get reliable risk estimates.

Risk validation by comparison with observed data is impossible, since risk is a composite figure, built out of many potential

events each with a different probability and consequences (i.e. risk cannot be monitored in the field). However, risk models' outcomes should not be used without considering their reliability for different uses. In fact, there are steps that can be taken to validate risk models' outcomes and to get a sense of their reliability; indeed, although risk cannot be monitored, its components can.

As explained by Sayers et al. (2016) a top-down or a bottom-up approach can be used. The top-down approach is often the

first obvious step. This approach starts by looking at risk models' results and compare these with experts' expectations, and use common sense in order to see if they look likely or if they can be falsified. This can, for example, be done by translating the risk figure in the damage which would be expected once in 10, in 100 or in 1000 years, and comparing that with observed damages in the flood-prone area, for similar historic events with known return periods. Based on this, experts can judge whether the risk estimates seem plausible.

An example of this common sense reasoning is described by Penning-Rowsell (2015) who compared outcomes of national-scale economic risk assessments for the UK with historic events damage figure, insurance claims and outcomes of earlier research. He concluded, based on comparisons with previous research and common sense, that the flood risk estimates were significantly overestimated, which may have large consequences for the efficiency of flood risk reduction investments in the UK. With his paper he provoked not only discussion but also further investigation into the usability of outcomes, uncertainties

and validation questions. Next, Sayers et al. (2016) explained the aim and usability of the risk outcomes in the UK. He made clear that risk outcomes on national scale can be confident even if local values of national scale analyses are wrong as it is expected that local errors counteract each other in such a way that the national estimate is sufficiently reliable.

The bottom-up approach can be used in the following step: all risk components could be assessed, their accuracy or validity evaluated and these assessments together can result in the validation of the resulting risk outcome. For example, one can look

at the uncertainties and reliability of models related to the various risk components and at how these propagate through the modelling chain and affect risk estimates; based on these uncertainty analysis, a reliability bound can be given.

When combining the validation outcomes of the risk components into an assessment of the validity of the total risk outcome, the validity of the combination should also be considered. To do so, it is crucial to understand which is the outcome of the models implemented for the different risk components. E.g. if a probability of 1/100 per year is provided as an outcome of a

dike safety model it is crucial to understand whether this relates to the exceedance probability of the design water level of the embankment, the probability of failure defined by geotechnical engineers (as "the probability that the embankment does not resist the water load anymore and some water is entering the protected area") or if it relates to the probability of a breach occurring. This may result in errors when combining this probability with consequences of failure.



A third option for risk models validation is comparing the risk model's outcomes with risk models' outcomes of different studies for the same area, or similar studies for similar areas, and explain the differences. Although it is not clear which model or study is to be preferred, such comparisons yield information on potential errors and uncertainties. It also may support insight in the required detail to gain accurate results. This benchmarking step is common practice, for example, in the insurance

industry. When flood risk and its reliability are estimated, then these can be communicated to the interested parties and the public. Ideally, risk estimates are communicated along with, and in such a way to represent, their reliability in a clear fashion. For example, they can be mapped in colours, or put in graphs or expressed in numeric figures with a certain band with. In any case, what it is important to communicate is analysts' trust in models' outcomes. If risk is expressed very precisely (e.g. 10.984 million euro per year), then users will have the tendency to consider it to be very accurate. However, if it is expressed roughly

(e.g. as 8 to 12 million euro per year), users may tend to test the acceptability of their decisions for risk equal to 8, 10 or 12 million euros.

## 4 Research and policy priorities

Validation is perhaps the least practised activity in current flood risk research and flood risk assessment. However, given the increasing role of flood risk assessment in policymaking, validation should become common practice. The two meetings that

this paper reports on highlight that there is a demand from both public and private decision makers on a better understanding of the reliability and uncertainties related to flood risk estimates. In this section, we provide some possible ways forward.

Good practices in model validation should be documented and communicated in the research and practitioners' communities, as benchmark solutions to be followed in contexts characterised by different physical features of the phenomena and the area under investigation, and different availability of data for validation. Some of these best practices are provided throughout this

paper. These include model-observation comparison, model-model comparison, model transfers, as well as expert judgement. A community of practice could be established to further promote the validation of risk models, potentially broadened beyond flood risk. Cross-hazard learning, including earth-quake and windstorms could be considered.

The collection of high quality data, especially in the field of defence failure and damage should be improved. The development of a platform to share data for flood risk model validation (for all the risk chain subcomponents) is highly desired. Collection

of high quality data on hazard, failure and impacts after events should be promoted, standardized and become common practice after major flood events, through research institutes and national organizations supported by national level policymakers. Untraditional tools for data collection (e.g. crowd sourcing, voluntary information approaches, social networks) should also be promoted to compensate unavailability of data, usually linked with unsustainable costs or technical unfeasibility of ad hoc collection campaigns.

An increased participation of end-users (i.e. public and private decision makers) into the research debate on flood risk models validation should be finally promoted, in order to tailor validation practices towards real needs. The commitment of end users into both the community of practice and the data share platform discussed before is highly desirable.



## 5 Conclusions

This paper presents and discusses the main results from two scientific events on flood risk models validation. Three aspects have been discussed (i) what is validation in flood risk modelling and why it is required, (ii) what techniques for validation are available for flood hazard, defence failure, and damage models, and (iii) which are the research and policy priorities in order

to improve the reliability of present flood risk estimates. We conclude that validation should be considered as the evaluation of model reliability towards its intended use. Three types of validation techniques have been discussed: the comparison between observed data and predictions, the comparison among different models' outcomes and expert judgement. The degree of implementation of the different techniques is discussed, among the different components of the risk modelling chain (i.e. hazard, defence failure and damage modelling).

This review reveals that, for all components, the paucity of observational data is the main constraint to model validation so that reliability of flood risk models can hardly be assessed.

We mentioned three recommendations. The collection of high quality data after every flood should be promoted, along with the development of a platform to share available/collected data. Last research efforts in these directions are discussed in the paper.  We also recommend to create a community of practice to promote and widespread best practices of flood risk models

validation. To improve validation and tailor it towards real needs we belief a greater commitment of end users in the research debate on validation in flood risk modelling is required

The path towards the accomplishment of these three main objectives will be the topic of a further forthcoming workshop on the topic of flood model validation that will be held at Politecnico di Milano (Milan, Italy) on 20-21 November 2017, entitled "Validation in Flood Risk Modelling: combining scientific, policy and market perspectives" (for more information see

www.eko.polimi.it).

**Competing interests**

The authors declare that they have no conflict of interest.

**Acknowledgments**

The authors acknowledge with gratitude all the participants and the speakers at the workshop in Delft (December 2014) and at

the special session at FloodRisk2016 (October 2016); as the main content of the papers was derived from the information presented and the fruitful discussions at these two events. We also acknowledge the European Geoscience Union (EGU), the Panta Rhei Initiative, the FloodRisk2016 Scientific Committee and Deltares for supporting these events.




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
