# Peer review of "Review Article: Validation of flood risk models: current practice and innovations"

_Natural Hazards and Earth System Sciences, 2017_

## Referee Comment (RC1) · Anonymous Referee #1 · 27 Oct 2017

General comments

The paper addresses a very important topic in flood risk assessment: the validation of the models. It is generally well organized and clear. The effort of the authors in collecting so many references is considerable. The paper could provide a substantial contribution to the understanding of the topic and a helpful guide to the reader in such a specific field, if the literature review was extended and summarized in more structured manner also by means of tables. Moreover, not all the flood risk model components are analyzed. Based also on what the authors wrote, the paper at present status is not precisely a review article, but rather a summary of the outcomes of two important workshops on the validation of flood risk models. On the one hand, it provides useful references in the literature; on the other hand, it gives extensive space to recommendations and authors' opinions or suggest priorities in the flood risk research and policy. This approach, although useful, is not usually adopted within a review article. Finally, often the problem of validation is assimilated in the paper to the problem of uncertainty estimation, while I believe they are two distinct problems. The paper is recommended for publication in NHESS after some minor revisions. I tried also to provide some suggestions that the authors are free to accept or reject.

Specific comments

Title. I suggest taking out from the title "review article". Moreover, I suggest substituting "current practice and innovations" with "practices, lacks and recommendations"

P1 L16-17. The components of risk modelling are not only flood hazard, defence failure and damage analysis. I think there are others like spatial and temporal correlation estimation, exposure or exposed values estimation, uncertainty estimation, stochastic events generation. Also, flood risk modelling could not be limited to damage analysis, but in general extended to indirect losses and social impacts.

P1 L28 – P2 L9. It is not clear the aim of the paper. The aim of a review paper is to summarize the existing literature on the topic. Instead in this paragraph, authors says that the paper aims at summarized the findings of two workshops and provide recommendations. I think it should be clarified better the role of these two workshops in the context of the paper. Maybe the aim could be addressed to how the findings of the workshops contributed to the understating and knowledge of the topic.

P2 L10-13. The key questions reported here are really useful to guide the reader through the paper. I suggest that each of the following paragraph replies to each single question: 1) What is validation … -> paragraph 2 2) Which are the techniques … -> paragraph 3 3) Which are the priorities … -> paragraph 4

P2 L15 – 18. I think this paragraph starts from too far. I don't think it is necessary to go to the foundation of engineering science to define what is validation. Also, the

analogy with a product is little bit out of context. Validation is a consolidated concept in science and in applied science. Nonetheless, I find interesting to define more closely the concept of validation in the context of flood risk modelling. I suggest modifying the introduction of paragraph 2 with something closer to the focus of the paper.

P2 L28-32. I find a little bit confusing to link the concept of validation with that of uncertainty estimation. I think they are two different concepts and it is necessary to clarify better what the authors really mean.

P3 L9-22. As said before, I suggest moving this part of the paragraph 2 to the paragraph 3.

P3 L14. I find what is stated in the parenthesis ("risk cannot be observed") quite imprecise. Risk is the probability of losses. The "modelled quantity" is loss which is measurable.

P3 L15-16. The sentence is contradictory. How could exist "validated" models if they are validated by comparison with other models?

P3 L19-23. This part of the paragraph is very useful to guide the reader into the different techniques described: i) objectives, ii) data, iii) techniques and iv) new developments. I suggest adopting those fields as the columns, and the flood risk model components as the rows, of a table that provides a general overview of the state of the art of flood risk validation as described in the paragraph 3.

Paragraph 3. There are very useful references in this paragraph. However, often the techniques are just mentioned but not explained or summarized. The benefit of this literature review would be to organize, to classify and to summarize the main contributions to the topic of flood risk model validation. Also, I think other important flood risk modelling components should be included. I think, the most relevant are: 1) the estimation of the temporal and spatial correlation of the flood events, 2) the stochastic generation of (synthetic) flood events, 3) the assessment of the exposure, 4) the

estimation of the uncertainty associated with the risk assessment.

P4 L29-33 and P6 L22-24. As in a previous comment, I found confusing to assimilate the model validation with the uncertainty estimation.

P6 L25-28. I think that the discussion on the uncertainty of the model or outputs is out of the scope of this paragraph.

P6 L28-29. I think that what is interesting here is how this type of models is validated.

P7 L2-4. I think the topic of the model transferability is out of the scope of this paragraph.

P7 L13-33. I think that the discussion on the lack of data and future research should be moved on the paragraph 4. I suggest focusing in paragraph 3 only on the review of the existing techniques.

P8 L4. I don't' agree with the statement "Risk validation by comparison with observed data is impossible", see comment above P3 L14.

---

## Referee Comment (RC2) · Anonymous Referee #2 · 30 Oct 2017

This paper addresses a very relevant topic, i.e. the validation of flood risk models. The manuscript is well written, and provides a nice review of methods. Thus, it has the potential to become an important paper. Yet, as pointed out by Reviewer #1, not all components of risk are considered. Moreover, the literature review is far from being comprehensive. Some key aspects (see below) are completely ignored.

My major concern is that, to validate these models, we must primarily assess their capability to capture "changes in flood risk" (which happens to be the name of the cited Panta Rhei Working Group!) across decades. However, the aspect of change over time is not sufficiently addressed by this manuscript. Assessing risk is not like measuring water levels. Flood risk is both real and socially constructed, e.g. see discussion about different definitions of risk in Section 2.2 of Hulbert and Gupta (2016). In flood risk

[Figure]

modeling, assessing the direction of change is perhaps more important that estimating potential losses, which are highly uncertain anyway given the unpredictability of indirect and intangible losses.

The vast majority of flood risk models struggle in capturing the direction of change. This is mainly because they cannot capture changes in vulnerability, and often ignore feed-back loops between social and hydrological components of risk. For instance, could any of these flood risk models capture the dynamics of flood losses in Bangladesh reported by Mechler and Bouwer (2015)? Moreover, flood risk assessments are often carried out to explore the effects of risk reduction measures. Can any of these models simulate the fact that structural protection measures reducing flood hazard often trigger a (more than expected) increase of exposure and vulnerability, as widely shown by the safe-development paradox (Kates et al., 2006)and shown with the catastrophic flood-ing events in New Orleans (2005) and Brisbane (2011)? What is the point of getting a precise and accurate number for a thing (risk) that is (at least partially) socially con-structed, while being unable to assess whether risk will actually increase or decrease a decade after the introduction of flood protection measures?

I presume that the authors are aware that various models of human-flood interactions have been developed by several socio-hydrologists over the past five years. Some of these models have given promising results in capturing the direction of flood risk changes. Yet, they are completely ignored in this manuscript. Thus, I suggest a major revision of the manuscript to address the fundamental issue of how well flood risk models capture changes over decades.

References

Kates, R.W., Colten C.E., Laska S., Leatherman S.P. (2006). Reconstruction of New Orleans after Hurricane Katrina: A research perspective. Proceedings of the National Academy of Science, 103(40):14653-14660.

Hurlbert, M. and Gupta, J. (2016). Adaptive Governance, Uncertainty, and Risk: Policy

[Figure]

Framing and Responses to Climate Change, Drought, and Flood. Risk Analysis, 36: 339–356.

Mechler, R., & Bouwer, L. M. (2015). Understanding trends and projections of disaster losses and climate change: is vulnerability the missing link? Climatic Change, 133(1): 23-35.
* * *

---

## Author Comment (AC1) · 8 Dec 2017

We thank the referees for their very useful suggestions and concerns which we'll try to mostly address in the new version of the paper in order to increase its robustness and its clearness for the reader. In the attachment, a point by point answer to the review is supplied.

Please also note the supplement to this comment:
https://www.nat-hazards-earth-syst-sci-discuss.net/nhess-2017-303/nhess-2017-303-AC1-supplement.pdf

2017-303, 2017.

---

## Author Comment (AC2) · 8 Dec 2017

We thank the referees for their very useful suggestions and concerns which we'll try to mostly address in the new version of the paper in order to increase its robustness and its clearness for the reader. In the following, a point by point answer to the review is supplied.

**Reviewer 1**

The referee raises five major concerns that we will try to address and some minor suggestions that we'll partly consider.

Major concerns

1) The paper is not a review → "the aim could be addressed to how the findings of the workshops contributed to the understanding and knowledge of the topic".

   We agree with the referee, as the paper actually reports the point of view of a meeting of 50 experts attending the two workshop events on flood risk models validation, and is far from being a complete review on the topic but more a critical analysis of it. "Review article" was initially added to the title in order to classify the paper among the types of manuscript accepted in NHESS. Reading again the description (https://www.natural-hazards-and-earth-system-sciences.net/about/manuscript_types.html) we think that the paper can be better classified as "invited perspective". After consultation with the executive editor Dr. Heidi Kreibich, we all agreed to change the manuscript type into "invited perspective".

   This type of manuscript must include "the author's perspective, (critical) observation, or research suggestions based on sound arguments, facts, published research studies, or real-life examples" on a topical aspect of natural hazards, that is just what our paper do with respect to flood risk model validation.

   For these reasons, we do not think that a full review of the literature is required.

2) often the problem of validation is assimilated in the paper to the problem of uncertainty estimation, while I believe they are two distinct problems

   The Referee is right. We consider the problems as equivalent because, according to our definition of validation, uncertainty assessment is included in validation.
   During the workshop several definitions of validation were discussed. However, we all agree that validation in flood risk modelling cannot be "strictly" intended as the check of model outcomes against recorded measures (as it is traditionally done). It should aim "at assessing whether risk assessments meet the needs of relevant stakeholders" or "at evaluating model reliability towards its intended use" (as we state in the paper).
   Then validation implies two main things:
   - that the model supplies an estimation of the variable(s) of interest (i.e. information relevant for stakeholder/decision maker)
   - the such estimation is supplied with a level of accuracy/precisions that is in line with the intended use of the estimate

   As the second point illustrates, validation is very closely related to uncertainty assessment.

   We understand that our viewpoint can lead confusion in the readers, and an explanation will be added into the revised version of the paper

3) not all the flood risk model components are analysed → "The components of risk modelling are not only flood hazard, defence failure and damage analysis. I think there are others like spatial and temporal correlation estimation, exposure or exposed values estimation, uncertainty estimation,

stochastic events generation. Also, flood risk modelling could not be limited to damage analysis, but in general extended to indirect losses and social impacts" → "I think other important flood risk modelling components should be included. I think, the most relevant are: 1) the estimation of the temporal and spatial correlation of the flood events, 2) the stochastic generation of (synthetic) flood events, 3) the assessment of the exposure, 4) the estimation of the uncertainty associated with the risk assessment"

a) In our opinion the definition of a "component" depends on the level of detail adopted in the analysis. Our 3 macro-components include all the (sub-)components recalled by the referee (but we would like to note that even such sub-components could be further divided in sub-components, e.g. one could distinguish between the assessment of physical and economic exposure, and so on).

For example, "The estimation of the temporal and spatial correlation of the flood events" and "the stochastic generation of (synthetic) flood events" are included in what we mean as "the assessment of the likelihood and the features of the damaging physical event". We also refer to them, i.e., "the uncertainty is usually addressed by adopting different flow hydrographs as upstream boundary conditions or by doing Monte Carlo simulations". Equally "the assessment of exposure" is included in "damage assessment" because a damage model will always consider exposure and vulnerability while, according to point 2), uncertainty estimation is part of the validation process and is discussed in the paper. In fact, also "defence failure assessment" can be included in hazard models but we decided to pay specific attention to this crucial and often neglected topic.
We are aware that each "sub-component" would require a specific investigation but we only report what emerged as crucial and primary during the debates.

For these reasons, we would not include all the subcomponents, but we will better justify our choices for the three components in the revised version of the paper, and explain that various sub-components are included.

b) We are aware that we neglect indirect loss and social impacts although we recognize their importance in the overall damage figure. The choice of focusing on direct damage is due to the evidence that much more models exist for direct damage estimation than for indirect or social impacts; nonetheless, even in the presence of more models, and then more knowledge, validation is difficult for many reasons (lacking of data, lacking of expertise, etc.). In such a context, we think that the debate must start from most known "fields" of research and expertise, to further extend the discussion in the future to other types of losses and impacts from floods. This will be explained in the new version of the paper.

4) what is stated in the parenthesis ("risk cannot be observed") is quite imprecise. Risk is the probability of losses. The "modelled quantity" is loss which is measurable.

We know that there is no consensus in the definition of risk (risk can be considered as an expected value, a probability distribution, the uncertainty of an event, etc.). In the paper risk is considered as the expected damage(loss) in a certain time period (usually damage/year) on a certain area, and integrates all event probabilities in that area. We cannot always measure the damage/loss associated to a certain event but not the risk (as often we have information on too little past events across the range of all event probabilities, and in highly protected areas damaging events hardly occur). Nonetheless, as stated by the referee, "flood risk modelling could not be limited to damage analysis, but in general extended to indirect losses and social impacts" which measurement is still controversial.
We think the paper already explains the point, i.e. "risk is a composite figure, built out of many potential events each with a different probability and consequences (i.e. risk cannot be monitored in the field)". A more detailed explanation can be added.

5) the literature review can be summarized in a more structured manner also by means of tables → P3 L19-23. This part of the paragraph is very useful to guide the reader into the different techniques described: i) objectives, ii) data, iii) techniques and iv) new developments. I suggest adopting those fields as the columns, and the flood risk model components as the rows, of a table that provides a general overview of the state of the art of flood risk validation as described in the paragraph 3.

This is a very nice suggestion. This effort can improve the readability of the paper and we'll add the table.

Minor suggestions

6) Title; substituting "current practice and innovations" with "practices, lacks and recommendations"

The title will be changed into "Invited perspective: Validation of flood risk models: current practice and possible improvements"

7) P2 L10-13. The key questions reported here are really useful to guide the reader through the paper. I suggest that each of the following paragraph replies to each single question: 1) What is validation : : : -> paragraph 2 2) Which are the techniques : : : -> paragraph 3 3) Which are the priorities : : : -> paragraph 4

We think the paper is already clear on this point.

8) I think this paragraph starts from too far. I don't think it is necessary to go to the foundation of engineering science to define what is validation. Also, the analogy with a product is little bit out of context. Validation is a consolidated concept in science and in applied science. Nonetheless, I find interesting to define more closely the concept of validation in the context of flood risk modelling. I suggest modifying the introduction of paragraph 2 with something closer to the focus of the paper.

In fact, the discussions we had at the workshops highlighted that validation is not so a "consolidated" concept in science but it can have different "nuances". A clear definition of what we mean with validation is required in the paper; this is just the authors' perspective we want to discuss. The reference to engineering science is due as readers can understand how the concept has been translated from its original context to flood risk modelling. We will not modify this part of the paper.

9) P3 L15-16. The sentence is contradictory. How could exist "validated" models if they are validated by comparison with other models

Here we refer to validated models for the same area (but for example working a larger scale so that they can be validated), or for similar areas. A sentence will be added to better explain the point.

10) P6 L28-29. I think that what is interesting here is how this type of models is validated.

As far as we know, the three approaches that can be used are always the same: comparison with observed data, comparison with validated models, expert judgments. However, a comment will be added on this.

11) P7 L13-33. I think that the discussion on the lack of data and future research should be moved on the paragraph 4. I suggest focusing in paragraph 3 only on the review on the existing techniques.

We agree with the referee and we will move the discussion as suggested.

**Reviewer 2**

The referee raises only one main concern (but for the request of extending the literature review – see comment 1 above)

The aspect of change over time is not sufficiently addressed by this manuscript … to validate these models, we must primarily assess their capability to capture "changes in flood risk".

For instance, could any of these flood risk models capture the dynamics of flood losses in Bangladesh reported by Mechler and Bouwer (2015)? Moreover, flood risk assessments are often carried out to explore the effects of risk reduction measures. Can any of these models simulate the fact that structural protection measures reducing flood hazard often trigger a (more than expected) increase of exposure and vulnerability, as widely shown by the safe-development paradox and shown with the catastrophic flooding events in New Orleans (2005) and Brisbane (2011)? What is the point of getting a precise and accurate number for a thing (risk) that is (at least partially) socially constructed, while being unable to assess whether risk will actually increase or decrease a decade after the introduction of flood protection measures?

We really thank the referee for arising this point that we neglected mainly because it was not discussed at the two workshops the paper refers to. However, we think that the steady nature of many flood risk models (i.e. hazard and damage models), the reliability of the estimate they provide and their validation towards dynamic systems (and then "dynamic" observations) must be mentioned and briefly discussed in the paper. This will be added to the revised version.

From another perspective, also the capacity of simulating "changes in flood risk", after the introduction of a certain mitigation measure is relevant for model reliability. Still, in our opinion, this is more related to the capacity of the modeller of taking into account, in the estimation of the "new" value of risk, the effects that such a measure has in terms of hazard and exposure/vulnerability scenarios, rather than to the model itself. This is what is usually done in cost-benefit analysis. In this case, validation of flood risk models is still more challenging as we should validate them with the "observed risk" before and after the introduction of the measure. Some comments will be added at the paper with respect to this.